# Knowledge, Attitude, and Perception of Health Care Providers Providing Medication Therapy Management (MTM) Services to Older Adults in Saudi Arabia

**DOI:** 10.3390/healthcare11222936

**Published:** 2023-11-10

**Authors:** Fawaz M. Alotaibi, Zainab M. Bukhamsin, Alanoud Nasser Alsharafaa, Ibrahim M. Asiri, Sawsan M. Kurdi, Dhafer M. Alshayban, Mohammed M. Alsultan, Bassem A. Almalki, Wafa Ali Alzlaiq, Mansour M. Alotaibi

**Affiliations:** 1Pharmacy Practice Department, College of Clinical Pharmacy, Imam Abdulrahman Bin Faisal University, Dammam 34221, Saudi Arabia; 2180002400@iau.edu.sa (Z.M.B.); 2210040168@iau.edu.sa (A.N.A.); imasiri@iau.edu.sa (I.M.A.); smkurdi@iau.edu.sa (S.M.K.); dmalshayban@iau.edu.sa (D.M.A.); mmaalsultan@iau.edu.sa (M.M.A.); baalmalki@iau.edu.sa (B.A.A.);; 2Pharmacy Practice Department, College of Clinical Pharmacy, King Faisal University, Al-Ahsa 31982, Saudi Arabia; mmqalotaibi@kfu.edu.sa

**Keywords:** medication therapy management, geriatrics, medication optimization, medication reconciliation

## Abstract

Introduction: Medication Therapy Management (MTM) is identified as a group of services provided to the patient in order to optimize the medication use in order to mitigate adverse drug reactions (ADRs), drug–drug interaction (DDI), and polypharmacy. Elderly populations above 60 years old are at high risk for Medication-related Problems (MRPs) due to several factors. Therefore, MTM programs showed good contributions globally regarding enhancing medication use in the elderly population. Thus, evident information regarding its implementation in Saudi Arabia is lacking in the literature. Objective: Our objective is to assess community pharmacists’ knowledge, attitude, and barriers to providing MTM services to the older adult population in Saudi Arabia. Methodology: A cross-sectional study has been conducted among community pharmacists across the Kingdom. It was survey-based research that was designed and conducted through (QuestionPro). The survey was distributed for the community pharmacists from Feb–May 2023 via (QuestionPro). Descriptive analysis was performed using SAS OnDemand to analyze the categorical variables and test it with the outcome of interest. Results: Out of the 528 participants who have viewed our questionnaire, 319 participants have completed the survey in 5 min average time. Most of our participants were male, holding a bachelor’s degree, and had an average working load of more than 40 h a week, respectively (84.95%, 92.48%, and 76.18%). In addition, the participants were from different regions of the Kingdom, which enhanced the generalizability of our findings. Moreover, 65.52% have reported a higher level of knowledge, while 34.48% have reported a moderate to low level of knowledge regarding MTM service. Most of those with a higher level of knowledge maintain a positive attitude regarding MTM service, its implementation, and dealing with older adult patients in the community pharmacy. In addition, lacking the time, training, and presence of a private consultation room were the top barriers to provide MTM services in the community pharmacy in Saudi Arabia. Conclusion: Educational sessions regarding MTM services among the older adult population are highly recommended for community pharmacists before its implementation.

## 1. Introduction

Medication Therapy Management (MTM) services are defined as “A distinct service or group of services that optimize therapeutic outcomes for individual patients” [1]. These services encompass a wide array of professional tasks and responsibilities that fall within the purview of licensed pharmacists or other qualified healthcare providers. The term “Medication Therapy Management (MTM)” was firstly introduced in 2003 following the enactment of the Improvement and Modernization Act (MMA) during President George Washington’s tenure. The initial implementation of MTM programs took place by Medicare service centers in the United States [2]. In July 2004, the American Pharmacists Association (APhA) and the National Association of Chain Drug Stores (NACDS) collaborated to establish a framework for the implementation of MTM services. This framework was designed to improve the efficiency of pharmacy practice in community settings. In addition, an advisory panel consisting of experts in healthcare settings then devised two iterations of the MTM framework model known as the “Core Elements of an MTM Service” in both version 1.0 and version 2.0 [3].

To achieve a successful MTM service, it is essential to fulfill a set of five core components to guarantee a favorable result. These essential elements encompass medication therapy review (MTR), personal medication record (PMR), medication-related action plan (MAP), intervention and/or referral, and finally, documentation and follow-up. Detailed explanations of each stage of the MTM service can be found elsewhere [3]. Nonetheless, the implementation of MTM services has proven to play a significant role in achieving cost savings and effectively identifying and addressing medication-related problems (MRPs) across various clinical scenarios, leading to heightened patient satisfaction. Supporting the MTM program can have a positive impact on overall healthcare cost reduction. As per 2001 data related to patients with hyperlipidemia and hypertension, the total personal expenses in the United States decreased by 31.5%, while prescription drug costs increased by 19.7% [1]. Furthermore, the overall cost of healthcare observed a substantial reduction of USD 3678 per individual annually, yielding a return on investment of USD 12.15 for each dollar invested in MTMs [1]. adverse drug reactions (ADRs) identified through MTM consultations pertain to medication appropriateness, efficacy, safety, and adherence [4]. To provide additional insight, a study carried out in 2021 uncovered 192 Medication-related Problems (MDRs) during MTM interventions for a sample of 384 individuals with type 2 diabetes, hypertension, and similar coexisting health conditions [5]. Notably, this study observed a notable increase in health promotion, particularly within the diabetic subgroup [5]. An additional study, which highlighted the beneficial impacts of MTM services, examined 63 patients dealing with asthma and chronic obstructive pulmonary disease, with the majority of cases involving complex medication regimens (53.2%). The MTMs offered to these patients produced favorable results, including the enhancement of medication administration techniques, treatment success, and an increase in medication adherence. Similarly, in Saudi Arabia, it is imperative to prevent adverse drug reactions (ADRs) in order to safeguard patient safety [6]. A study examining 26,808 medication orders in KSA revealed that the identified issues predominantly revolved around prescribing errors, dispensing errors, and administration errors [6].

Nonetheless, Saudi Arabia promotes the adoption of MTM services, with the Saudi Society of Clinical Pharmacy (SSCP) actively endorsing the acquisition of comprehensive medication therapy management expertise by clinical pharmacists [7]. However, research concerning MTM and its influence on hospitals or community pharmacies remains insufficient within the Saudi context. A cross-sectional study aimed to assess the attitudes, intentions, and knowledge pertaining to the delivery of MTM services in Saudi Arabia showed a positive attitude regarding MTM with a strong intention, high pressure to provide the service, strong control, and good knowledge about MTM services. Furthermore, the most favorable outcomes and proficiency were observed among pharmacists who had successfully finished residency programs [7]. Moreover, the majority of the sample agreed that the clinical pharmacist is the best person to provide MTMs and considered that the current workload is the main barrier during providing MTM services, and (96%) enjoyed the experience of providing MTM service [8].

As previously noted, the development of MTM services in Saudi Arabia is still in its nascent stages, and assessing their effectiveness has become imperative for policymakers and stakeholders [7]. This assessment is particularly crucial concerning MTM services for the elderly population, who experience a higher prevalence of medical conditions, polypharmacy, and an increased likelihood of reporting adverse drug reactions (ADRs). It is noteworthy that the elderly demographic in Saudi Arabia is expanding, with a projected increase to 18% in 2050, a substantial rise from the current 3.5% [9]. Older adult population are in high risk of developing ADRs due to several factors such as psychological changes, physiological changes, comorbidities, and polypharmacy [8,10,11,12]. As a results, one study in Saudi Arabia reported 111 medication-related problems, and in another study, the prevalence of drug–drug interactions among geriatric population was over 90% of cases [13,14]. Interestingly, polypharmacy was a key player in causing ADRs, DDI and other negative consequences of using necessary medications [15,16]. As a results, MTM programs have shown a positive contribution regarding enhancing medication optimization in geriatric population [17,18,19,20,21].

To better understand the future impact of the pharmacist on the MTM program workflow, further evaluation of the knowledge, attitude, and practice can be performed to promote the outcomes of the medication use process in KSA and potentially enhance the care in various populations. To our knowledge, the MTM programs impact on the elderly is not adequately assessed in KSA in the community settings. Therefore, our objective is to assess pharmacist knowledge, attitude, and barriers to providing medication therapy management services to the geriatric population in Saudi Arabia.

## 2. Methods

### 2.1. Study Design

This is a cross-sectional study, with an online survey-based design from the period of February–May 2023.

### 2.2. Ethical Considerations

Participants had to sign a consent form in order to be recruited. The participation was voluntary and completely anonymous. The ethical approval was granted by the Institutional Review Board of Imam Abdulrahman Bin Faisal University (UGS-2022-05-522 on 12 December 2022).

### 2.3. Setting

This study focused on interviewing community pharmacists across the Kingdom of Saudi Arabia who serve the older adult population directly or indirectly.

### 2.4. Sampling Technique

The estimated number of community pharmacists in Saudi Arabia is around 20,000 by using the following formula: Sample Size (n) = [(Z^2^) ∗ (*p*) ∗ (1 − *p*)]/E^2^). The sample size required to achieve a 95% confidence interval with a 5% margin of error for a population of around 20,000, which is equal to 385.

### 2.5. Population

Community pharmacists were recruited from different community pharmacy chains in the KSA. Community pharmacists interested in caring about the older adult population or who have previously worked with older adults were invited to participate in the study. A clear statement was made at the beginning of the survey asking the pharmacists if they are specialized in geriatrics, have interest in working with older adults, or have previously worked with older adults as a requirement to start the survey, otherwise they are not eligible to continue the study. Students, pharmacist interns, and pharmacy technicians were excluded from the study. In addition, any other community pharmacists who are not working closely with older adults or have no interest in working with this population were excluded from the study. Finally, the survey was written in English; therefore, only participants who could read and write in English could complete the survey.

### 2.6. Data Collection

The ethical approval was granted prior to starting data collection. The data were collected using an online self-report questionnaire from the period of (February–May 2023). The research assistants visited several community pharmacies to introduce the research idea and to increase the response rate at the beginning of the study. Furthermore, we have asked the community pharmacists who work in some of the chain community pharmacy to distribute the survey link to their colleagues who are eligible for this study via the study’s different social media platforms like work emails and WhatsApp to enhance the number of responses. The previous mentioned platforms were used as well to remind the participants to complete the survey. The data collection form was developed via QuestionPro, a trusted platform widely used in questionnaire-based research.

### 2.7. Study Tool

The research tool was adopted from a validated questionnaire developed by Al-Tameemi et al. [20]. A few minor modifications were applied to the validated questionnaire to achieve the aim of this study. After that, the questionnaire was reviewed by experts in the field and piloted by 10 community pharmacists to ensure its validity. No changes were made after the piolet study was conducted. The Cronbach alpha was 0.75, which is a good indication of the survey reliability. The survey encompassed of four sections as follows: demographic and general questions (e.g., age, gender, geographic area, years of experience, availability of other pharmacy staff, and workload); knowledge-based questions regarding MTM, which includes 5 (true/false/I don’t know) questions. We purposely created the “I don’t know” option to prevent guessing choices in the test. All 5 questions test the core elements of the MTM services and its objectives, goals, and outcomes. Each question was worth one point; thus, the maximum score was 5 marks for each participant. Then, we categorized the respondents into high level of knowledge (score = 5), moderate knowledge (score = 3 and above) and low knowledge level (score = below 3). In the questionnaire, we asked 20 questions to test the pharmacists’ attitude regarding providing MTM services in general and among older adults population. In addition, we have asked several questions to test the potential barriers that could prevent conducting an MTM in the community pharmacy. All the distributed questions are available in the Appendix A provided.

### 2.8. Statistical Analysis

The data were analyzed using SAS OnDemand software, https://welcome.oda.sas.com/. The mean and standard deviation were reported for the continuous variables as well as frequency and percentage for categorical variables. For the sake of simplicity, we have combined the “Strongly Agree and Agree responses” and “Strongly Disagree and Disagree responses” into two separate groups. On the other hand, each question under the “Knowledge” variable was given a single score, and the sum of the scores determined the level of knowledge (i.e., high and moderate to low levels of knowledge). The chi-square test was applied to test the differences between the high-level knowledge group compared to moderate- to low-level knowledge. All the variables were tested at a 0.05 level of significance.

## 3. Results

### 3.1. Respondent Characteristics

A total of 528 participants viewed the questionnaire, and 319 participants completed the survey. Table 1 displays the frequency of participants according to the demographic and sociodemographic characteristics. The majority of our study participants were male (n = 271, 85%) and in the age group 30–39 years old (n = 174, 55%). The majority of pharmacists held bachelor’s degrees (n = 295, 92%). Moreover, most pharmacists had a working load of more than 40 h/week (n = 243, 76%) compared with pharmacists with 40 h or less/week (n = 76, 24%). In addition, the participants were equally distributed from different regions of the Kingdom.

### 3.2. Knowledge

Most of the community pharmacists in our study (66%) (n = 209) showed a high level of knowledge of MTM services by scoring 5 out of 5 in the knowledge part. On the other hand, the remaining community pharmacists (34%) (n = 110) showed a moderate or low level of knowledge by scoring four or less.

### 3.3. Attitude and Perception

As shown in Table 2, the attitude and perception regarding MTM services in Saudi Arabia, in general, were associated with the level of knowledge. The community pharmacists with moderate or low levels of knowledge (n = 110) showed a neutral attitude and perception regarding MTM implementation, online education, or even life workshops. The following results exclusively represent the sample with the high level of knowledge (n = 209). A majority (n = 180) showed a positive attitude regarding becoming an MTM provider if these services were implemented in the future. In addition, 70% of the high-level knowledge pharmacists agreed that MTM services are hospital-based programs, while 30% had a moderate to low level of knowledge. Moreover, 80% of those with a high level of knowledge believed that online education is a good way to provide training about MTM, compared to 20% with a moderate or low level of knowledge. Lastly, 80% of the high-level knowledge pharmacists anticipate having enough time to apply MTM services in the future compared to 20% with moderate or low levels of knowledge.

In addition, 60% of moderate to low level of knowledge group strongly disagree or disagree that they will have enough time to provide MTM services compared to those of high level knowledge, who agree that they will have enough time to provide MTM services (70% of the participants with high level knowledge). The presence of a private consultation room could be a potential barrier due to most participants acknowledging that their pharmacy does not provide a private consultation room—a necessary room to provide MTM services (Table 2). As shown in Table 3, we asked about training as a potential barrier, which 82% of the high level of knowledge group agreed that it is a barrier for providing MTM services.

Table 3 presents the differences between high-knowledge pharmacists and the moderate to low level of knowledge group regarding the attitude and perception of providing MTM services to older adults. The sample with a moderate or low level of knowledge (n = 110) showed a neutral attitude and perception in all 14 questions that were specifically for MTM regarding older adults, which reflects that the level of knowledge does impact the pharmacist attitude and perception regarding MTM services provided to the elderly. The following results exclusively represent the sample with the high level of knowledge (n = 209), with a majority (n = 164) believing that an elderly-directed MTM program subscription is worthy of initiation. A majority (n = 184, 80%) of the high level of knowledge group believed that implementing elderly-directed MTM services will improve their quality of life. The community pharmacists with a high level of knowledge (n = 165, 80%) were interested in learning more about elderly-directed MTM service compared to 20% of those with moderate to low levels of knowledge. Overall, 75% (n = 120) of high-level knowledge pharmacists were willing to provide MTM services by personal visits to the elderly homes/centers compared to only 25% of moderate to low level of knowledge who were willing do so. Regarding accessing geriatrics guidelines and getting familiar with them, 73% of those with a high level of knowledge agreed or strongly agreed with it compared to only 27% with a moderate to low level of knowledge. The rest of the questions related to the attitude and perception of community pharmacists regarding MTM service to the older adult population can be found in Table 3.

## 4. Discussion

To the best of our knowledge, this is the first study that investigated the knowledge, perception, and attitude/barriers of community pharmacists in Saudi Arabia regarding MTM services. Although more than half 65% of the study participants showed a high level of knowledge, 34% of community pharmacists showed a moderate to low level of knowledge regarding an important service that community pharmacists could provide to the older adult population to prevent ADRs and enhance their quality of life [21,22,23,24]. This percentage could be attributable to PharmD curriculum not focusing on MTM programs, especially among geriatrics population or chain community pharmacies that particular services are not of their interest at this stage of time.

In a comprehensive systematic review of the pharmacists knowledge regarding MTM services, the author found that 12 papers out of 17 had a considerable amount of knowledge on MTM services or had some knowledge regarding MTM core elements; these results are similar to what we have found in our study, suggesting that pharmacists are a good fit for such services [17,21]. Moreover, one study found that there was a positive public perception of community pharmacists, leading the pharmaceutical care service to the patient, which will help us with MTM implementation in the future [25]. In the same systematic review study, they have suggested that pharmacists with higher knowledge are more likely to have a positive attitude regarding MTM service implementation, similar to what we have found, with 78% of those with higher knowledge willing to provide MTM services in the future compared to 22% of those with low level of knowledge. This gives us hope that increasing pharmacists’ knowledge by providing continuous workshop related to MTM implementation and its skills will help ease the process of its implementation.

On the other hand, we have studied the barriers that the community pharmacists may face or think will prevent them from providing such services. One of the major barriers we found in our study was the lack of training in MTM services for community pharmacists, which was similar to the previous studies [20,22]. Thus, the school of pharmacy and professional societies should focus on providing the appropriate training to pharmacists in order to master such services. Another suggestion was that all chain pharmacies in the Kingdom should start an initiative to provide a mandatory course for entry-level pharmacists regarding MTM services, which will enhance and ease the establishment of this important service in the future. Moreover, the Ministry of Health and insurance companies in Saudi Arabia are responsible for mandating MTM services among older adults to mitigate ADRs, DDIs, and increase the optimal therapy for this vulnerable population. In addition, time was a significant barrier to providing MTM services in the community pharmacy, which led us to suggest that making this service free of charge will encourage the pharmacists to start implementing MTM services. Another suggestion was that designated pharmacists focused on only providing MTM services will help to lessen the workload on the other pharmacists in the same pharmacy. Moreover, a designated private place in the community pharmacy is another barrier to implementing MTM services. Although, having a designated place to protect patient confidentiality is a part of Saudi pharmacy law, most community pharmacies are not using it in the way that it should be used. Timing is another issue reported by our participants regarding providing MTM services, like the previous studies [7,22].

In our study, we have found that most of our participants have a positive attitude regarding the older adult population and have a positive attitude regarding implementing MTM services to such a vulnerable population. This attitude will help the policymakers implement MTM services quickly and without any resistance from the pharmacists. We believe that such an attitude came from cultural, religious, and educational factors contributing to such a positive attitude. This attitude will help and facilitate the MTM implementation process in the future for the older adult populations in Saudi Arabia. In addition, our participants have positive confidence and competencies in terms of counselling older adult patients and delivering MTM services. This critical skill is needed to positively impact the MTM process.

It is suggested that pharmacy schools increase the content of MTM-related topics to different patient populations, like geriatrics in the curriculum. In addition, the Saudi Society of Clinical Pharmacists has started an initiative to educate pharmacists regarding MTM services, but this initiative is still under investigation. Another scientific society that can take the lead in this context is the Saudi Geriatric Society, a physician-based society. This society comprises many geriatricians who can enrich the scientific content and transfer their knowledge to pharmacists and other healthcare providers who are licensed to provide MTM services in Saudi Arabia.

This study has several strengths, which include the fact that the study participants were equally distributed to most of the Kingdom of Saudi Arabia’s region, which enhances the generalizability of the study’s findings. In addition, in collaboration with some of the community pharmacy chains in Saudi Arabia, we had easy access to community pharmacists, which impacts the data validity, taking into consideration the nature of the study design.

However, the results cannot be interpreted without considering some limitations. First, recall bias and selection bias due to the nature of the questionnaire-based study should be recognized. Second, most of our study participants were male (84%), which could limit the generalizability of our findings, but this could be explained by the fact that most female Saudi pharmacists preferred to work in hospital-based pharmacies and not community pharmacies, and only represent 10.8% of the total pharmacists who work in community settings according to a study conducted in 2021 compared to men who make up 89% of the total [24]. Third, although over 500 participants viewed the online questionnaire, we could not recruit more than 319 participants, a number that was less than what we calculated previously to generate a robust findings. Lastly, although we have asked in the questionnaire about the interest in working with the older adult population, we could not limit the participants to only those with specialization in geriatrics due to the following: 1. the limited number of clinical geriatricians in the Kingdom, and 2. most of the community pharmacists in Saudi Arabia are not specialized in any area of the field, and therefore, we have asked about their interest and willingness to work with older adults to overcome such limitations.

Future studies are needed to investigate the relationship between the willingness of providing MTM services and other predictors. Another important study suggested would involve focusing on studying the direct and indirect economic impact of implementing MTM services in the general population and among older adults, in particular in Saudi Arabia.

## 5. Conclusions

We conclude that medication therapy management services in Saudi Arabia are still not fully mature services that can be provided in community pharmacies, especially to the older adult population. Our study findings indicated that a moderate or low level of knowledge regarding MTM services is associated negatively with the willingness to provide the service to older adults. Therefore, educational training for community pharmacists is needed before its implementation. In addition, future studies are needed to assess the effectiveness of MTM service implementation nationwide among the older adult population. This will help the county achieve an essential pillar in the Health Transformation Plan 2025 in its Vision 2030.

## Figures and Tables

**Table 1 healthcare-11-02936-t001:** Baseline demographic and characteristics of community pharmacists, n = 319.

Characteristics	n (%)
Age	
20–29 years	120 (37.62)
30–39 years	174 (54.55)
40–49 years	24 (7.52)
50 Years or older	1 (0.31)
Gender	
Male	271 (84.95)
Female	48 (15.05)
Educational Level	
Bachelor’s Degree	295 (92.48)
Master’s Degree	4 (1.25)
PhD Degree	15 (4.70)
Residency or Post-bachelor’s Degree	5 (1.57)
Years of Experience	
Less than 2 Years	50 (15.67)
2–5 Years	67 (21.00)
6–10 Years	109 (34.17)
11–15 Years	72 (22.57)
More than 15 Years	21 (6.58)
Assistance Availability	
Yes	192 (61.15)
No	122 (38.85)
Total Number of Prescription Medication Prescribed for Older Adults/Day	
Less than 50 Rx	188 (58.93)
50–300 Rx	92 (28.84)
More than 300 Rx	14 (4.39)
Not Sure	25 (7.84)
Geographical Region	
Central Region	91 (28.53)
Eastern Region	76 (23.82)
Western Region	61 19.12)
Northern Region	46 (14.42)
Southern Region	45 (14.11)
Working Load/Week	
40 h or less/week	76 (23.82)
More than 40 h/week	243 (76.18)

Rx = Prescription medication.

**Table 2 healthcare-11-02936-t002:** Pharmacist attitude regarding providing medication therapy management services in general.

Question	Knowledge—High Level, n (%)	Knowledge—Moderate or Low Level, n (%)	*p*-Value
1. If MTM service will be implemented in the future, would you like to be an MTM service provider?			<0.0001
Strongly Agree or Agree	180 (77.59)	52 (22.41)	
Neutral	27 (32.93)	55 (67.07)	
Strongly Disagree or Disagree	2 (40.00)	3 (60.00)	
2. Do you think that providing MTM services is only hospital-based program?			<0.0001
Strongly Agree or Agree	84 (70.59)	35 (29.41)	
Neutral	44 (41.12)	63 (58.88)	
Strongly Disagree or Disagree	81 (87.10)	12 (12.90)	
3. Is online education a good way to provide training about MTM?			<0.0001
Strongly Agree or Agree	151 (80.75)	36 (19.25)	
Neutral	45 (38.79)	71 (61.21)	
Strongly Disagree or Disagree	13 (81.25)	3 (18.75)	
4. Do you prefer live workshops as a training method about MTM?			<0.0001
Strongly Agree or Agree	150 (78.95)	40 (21.05)	
Neutral	51 (44.35)	64 (55.65)	
Strongly Disagree or Disagree	8 (57.14)	6 (42.86)	
5. Do you think that you will have enough time to apply MTM service in the future?			<0.0001
Strongly Agree or Agree	138 (79.31)	36 (20.69)	
Neutral	63 (50.40)	62 (49.60)	
Strongly Disagree or Disagree	8 (40.00)	12 (60.00)	
6. Does your pharmacy or the place that you work at currently have a private consultation room for the patients?			<0.0001
Strongly Agree or Agree	118 (74.21)	41 (25.79)	
Neutral	35 (36.08)	62 (63.92)	
Strongly Disagree or Disagree	56 (88.89)	7 (11.11)	

*p*-value of <0.005 considered statistically significant using chi-square.

**Table 3 healthcare-11-02936-t003:** Pharmacist attitude regarding providing medication therapy management services to geriatric patients.

Question	Knowledge—High Level, n (%)	Knowledge—Moderate or Low Level, n (%)	*p*-Value
1. Do you think that elderly directed MTM services subscription programs are worthy of initiation in primary care?			<0.0001
Strongly Agree or Agree	164 (78.47)	45 (21.53)	
Neutral	41 (40.59)	60 (59.41)	
Strongly Disagree or Disagree	4 (44.44)	5 (55.56)	
2. Do you think implementing elderly directed MTM service in the future is important to improve their quality of life?			<0.0001
Strongly Agree or Agree	184 (80.00)	46 (20.00)	
Neutral	24 (28.24)	61 (71.76)	
Strongly Disagree or Disagree	1 (25.00)	3 (75.00)	
3. Are you interested in learning more information about elderly directed MTM service?			<0.0001
Strongly Agree or Agree	165 (79.71)	42 (20.29)	
Neutral	40 (38.10)	65 (61.90)	
Strongly Disagree or Disagree	4 (57.14)	3 (42.86)	
4. I am willing to provide phone counselling about MTM to the elderly patients?			<0.0001
Strongly Agree or Agree	131 (77.06)	93 (22.94)	
Neutral	67 (50.00)	67 (50.00)	
Strongly Disagree or Disagree	11 (73.33)	4 (26.67)	
5. I am willing to provide home visits to deliver the MTM services to the elderly centre’s/homes?			<0.0001
Strongly Agree or Agree	120 (75.47)	39 (24.53)	
Neutral	63 (50.40)	62 (49.60)	
Strongly Disagree or Disagree	26 (74.29)	9 (25.71)	
6. In your current practice, do you think that you spend enough time counselling your elderly patients?			<0.0001
Strongly Agree or Agree	144 (77.42)	42 (22.58)	
Neutral	52 (45.61)	62 (54.39)	
Strongly Disagree or Disagree	13 (68.42)	6 (31.58)	
7. Do you usually access (online or hard copies) most updated geriatric directed treatment guidelines available for diseases such as AGS Beers Criteria^®^ etc.?			<0.0001
Strongly Agree or Agree	116 (73.42)	42 (26.58)	
Neutral	61 (50.00)	61 (50.00)	
Strongly Disagree or Disagree	32 (82.05)	7 (17.95)	
8. Do you have an easy access (online or hard copies) to the geriatric guidelines to manage your elderly patients?			<0.0001
Strongly Agree or Agree	117 (78.52)	32 (21.48)	
Neutral	58 (45.67)	69 (54.33)	
Strongly Disagree or Disagree	34 (79.07)	9 (20.93)	
9. Lack of training in MTM services delivery is one of the potential barriers regarding applying elderly directed MTM service in the future?			<0.0001
Strongly Agree or Agree	156 (82.54)	33 (17.46)	
Neutral	49 (41.53)	69 (58.47)	
Strongly Disagree or Disagree	4 (33.33)	8 (66.67)	
10. Do you think that applying elderly directed MTM services need high budget?			<0.0001
Strongly Agree or Agree	122 (82.43)	26 (17.57)	
Neutral	69 (47.59)	76 (52.41)	
Strongly Disagree or Disagree	18 (69.23)	8 (30.77)	
11. I believe that geriatrics should receive more care than any other group of patients?			<0.0001
Strongly Agree or Agree	149 (83.71)	29 (16.29)	
Neutral	56 (43.41)	73 (56.59)	
Strongly Disagree or Disagree	4 (33.33)	8 (66.67)	
12. I feel confident in my communication skills while counselling an elderly patients?			<0.0001
Strongly Agree or Agree	153 (83.15)	31 (16.85)	
Neutral	49 (40.50)	72 (59.50)	
Strongly Disagree or Disagree	7 (50.00)	7 (50.00)	
13. I have the required competencies to deliver MTM services to the elderly patients?			<0.0001
Strongly Agree or Agree	150 (82.42)	32 (17.58)	
Neutral	53 (41.73)	74 (58.27)	
Strongly Disagree or Disagree	6 (60.00)	4 (40.00)	
14. I believe that geriatrics should receive more education than any other group of patients?			<0.0001
Strongly Agree or Agree	169 (80.48)	41 (19.52)	
Neutral	38 (37.25)	64 (62.57)	
Strongly Disagree or Disagree	2 (28.57)	5 (71.43)	

*p*-value of <0.005 considered statistically significant using chi-square.

## Data Availability

All the data of this research have been presented in this paper; however, the raw data are available upon request from the corresponding authors (Alotaibi FM).

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
