# Peer review of "Knowledge, Attitude, and Perception of Health Care Providers Providing Medication Therapy Management (MTM) Services to Older Adults in Saudi Arabia"

_healthcare, 2023, doi:10.3390/healthcare11222936_

Round 1

Reviewer 1 Report

Comments and Suggestions for Authors

The manuscript rises the extremely important issue of rationalization of pharmacotherapy aimed at improving the patients’ quality of life and saving costs of pharmacotherapy problems (such as duplication of therapies, potential adverse drug events, and variations between prescribers medications). Rationalization of phamacotherapy brings huge financial and health benefits in all areas of the medical system, and at all stages of the patient's life, especially for older patients.

 Nevertheless, the manuscript requires a revision to make it valuable material for publication, especially in methodological part.

 There is a need to indicate what the general population of pharmacists in Saudi Arabia is, and what part of this population are men. It is remarkably interesting why there is such an overrepresentation of men in the research - is there any data on the employment structure of women and men in Saudi Arabia pharmacy?

It is also unclear what was the method of filtering pharmacists working with the elderly. Was the information at the beginning of the survey that the questionnaire was addressed only to pharmacists working with the elderly and others should not complete it, or did pharmacists who do not work with people older people were excluded from the analysis after completing the questionnaire?

Were there any changes made to the research tool after the plotting?

The readers are not provided with clear information on the basis of what type of the pharmacists' knowledge was assessed, e.g., examples of questions could be presented. There is only a brief information that pharmacists’ knowledge was assessed by asking the questions, but it is unknown how many questions there were, and what exactly they concerned - the questionnaire should be included in supplementary materials.

What was the scale for assessing the knowledge and how many points indicated a high, low, or average level of knowledge. This is valuable information due to the fact, that the level of knowledge is the most important variable in the conducted analysis of pharmacists' attitudes in the presented research.

From a sociological point of view, part of the questions addressed to respondents and referring pharmacists’ opinions is suggesting in nature.  E.g., “Do you think that elderly directed MTM services subscription programs are worthy of initiation in primary care?” Do you think that applying elderly directed MTM services need high budget?”, or “Lack of training in MTM services delivery is one of the potential barriers regarding applying elderly directed MTM service in the future?

Regarding the barriers presented in the discussion part, the authors pointed out “lack of training” as one of the major barriers identified in the research. When reading the Results part, we cannot find any barriers indicated by the respondents. Does it mean that authors pointed to this barrier only based on the answer to the question mentioned above: “Lack of training in MTM services delivery is one of the potential barriers regarding applying elderly directed MTM service in the future? If does, it is far-reaching statement. The description of the study results did not indicate any other way of identifying barriers in implementing of MTM services.

 Additionally references need to be developed.

Author Response

All the reviewer comments were addressed in the attached file. Thank you so much for the valuable comments. 

Reviewer 2 Report

Comments and Suggestions for Authors

- Line 12; MTM full abbreviation should be mentioned.

- Line 13; ADRs, DDI full abbreviation should be mentioned.

- Line 18; knowledge, attitude, and barriers should be written in small letters.

- Lines 21-22; the sentence is not clear and need to be rephrased.

- Nothing mentioned about barriers in the abstract.

- Line 38; MTM full abbreviation should be mentioned.

- Lines 53; the words should be written in small letters.

- Line 106; knowledge, attitude, and barriers should be written in small letters.

- The study design section in line 109 should mentioned the type of survey study (online or paper-based) and the time frame for data collection. Move this information from "Data collection section".

- Sampling technique is not mentioned.

- Line 123; Pharmacist-interns should be written in small letters.

- Line 130; please clarify the sentence "Participants were mainly recruited through gatekeepers who helped disseminate the online questionnaire.".

- Mention the platforms used to distribute the link of the survey.

- Line 141; add the reference for the survey tool. Does it needs permissions?

- Line 143; I think that the words "how many" were directed from one author to another while drafting the manuscript and they did not add the number of experts who validated the survey.

- What are the outcomes of the piloting phase and which statistical analysis did you perform for it?

- No need to present the percentages using tw0 decimal points.

- The abbreviation for Rx should be mentioned in the footnote for Table 1.

- There is no details concerning the questions that measured the knowledge domain, which is very important for the manuscript. 

- Line 193; the sentence "directed to the elderly population is associated with the level of knowledge" is not accurate as chi-square test does not measure association.

- There is no information regarding the barriers in the results section despite that it is clearly mentioned in the title of the manuscript.

- The statistical analysis of the study is very simple with no inferential analysis that can enable the authors to draw robust conclusion.

- Line 216; there is no mention for barriers, which show that the authors did not focus on it.

- Line 233; "One of the major barriers we found in our study was the lack of training in MTM services for community pharmacists". It is the only sentence that talked about barriers in your manuscript. You did not examine aby other barriers.

- How did you measure attitude? it is only descriptive results with no tangible findings. How did you define positive attitude? as you mentioned in line 241 "most of our participants have a positive attitude"?

- There is no enough interpretation for the study findings in the discussion section, no enough comparison with previous relevant literature, and no enough citations for relevant studies.

Comments on the Quality of English Language

Needs extensive English language check.

Author Response

Thank you so much for the valuable comments, all the comments were addressed in the attached file. 

Reviewer 3 Report

Comments and Suggestions for Authors

Fawaz M. Alotaibi et al. submitted to Healthcare an article focusing to the knowledge, the attitude and the perception of Healthcare Workers providing MTM to geriatric population.

Below are the aspects that need to be clarified and implemented:

- please specify in the title that this is a study targeting Saudi Arabia;

- how many total Community Pharmacists was it possible to recruit? In other words, what would the “denominator” be? This aspect becomes essential to understand whether it is possible to apply statistical inference;

- the discussion must be implemented by comparing the results with other sector studies present in the biomedical literature;

- please conform the way references appear in the text, according to the Instructions for the Authors of the Journal; the same for the Author's Contributions section.

Comments on the Quality of English Language

Moderate editing of English language required

Author Response

(The authors gave the same response as above.)

Round 2

Reviewer 2 Report

Comments and Suggestions for Authors

The authors did not address all of my comments. Still, the analysis is not appropriate and descriptive and the discussion was not improved.

Author Response

In our study we focus on describing an important issue we face it every day in our community pharmacy and its documentation in a scientific study is crucial to our future health care transformation. We genuinely did not want to investigate the relationship between the outcome of interest and other possible predictors due to the scope of our research, and other administrative factors, timing, research assistant etc.

    In addition, we thoroughly reviewed the discussion and added our interpretations to it and we strongly believe it’s much better than the previous version.

Reviewer 3 Report

Comments and Suggestions for Authors

To apply statistical inference, you would have needed to enroll at least 385 respondents, whereas your study obtained 319 respondents. This means, in other words, that the sample is not representative of the entire population. This gap cannot be remedied with any round of refereeing.

Comments on the Quality of English Language

Moderate editing of English language required

Author Response

Absolutory right, although our online questionnaire was reached more than 600 possible participants and viewed and partially completed by more than 519 participants, we only included those who have completed 100% the survey (all sections and all possible variables), which were only 319. We have acknowledged that in our study limitations as well in the edited discussion.